# A comprehensive analysis of transcriptomic data for comparison of plants with different photosynthetic pathways in response to drought stress

**Shima Karami[1], Behrouz Shiran[1,2]\*, Rudabeh Ravash[1], Hossein Fallahi[3]**

1 Department of Biotechnology and Plant Breeding, Faculty of Agriculture, Shahrekord University, Shahrekord, Iran, 2 Institute of Biotechnology, Shahrekord University, Shahrekord, Iran, 3 Department of Biology, Faculty of Sciences, Razi University, Kermanshah, Iran

\* shiran@sku.ac.ir, beshiran45@gmail.com

**Data Availability Statement:** All relevant data are within the paper and its Supporting information files.

## Abstract

The main factor leading to a decrease in crop productivity is abiotic stresses, particularly drought. Plants with C4 and CAM photosynthesis are better adapted to drought-prone areas than C3 plants. Therefore, it is beneficial to compare the stress response of plants with different photosynthetic pathways. Since most crops are C3 and C4 plants, this study focused on conducting an RNA-seq meta-analysis to investigate and compare how C3 and C4 plants respond to drought stress at the gene expression level in their leaves. Additionally, the accuracy of the meta-analysis results was confirmed with RT-qPCR. Based on the functional enrichment and network analysis, hub genes related to ribosomal proteins and photosynthesis were found to play a potential role in stress response. Moreover, our findings suggest that the low abundant amino acid degradation pathway, possibly through providing ATP source for the TCA cycle, in both groups of plants and the activation of the OPPP pathway in C4 plants, through providing the electron source required by this plant, can help to improve drought stress tolerance.

## Introduction

The impact of water scarcity is already being experienced in several regions around the world, and climate change in the future can lead to the aggravation of the drought crisis. In addition to decreasing the frequency and amount of precipitation, rising global temperatures result in increased evapotranspiration and water loss. Therefore, it is expected that the amount of irrigation will increase in order to meet the growing needs of the crop, which will also lead to the depletion of groundwater. To address the expanding areas with water shortages, it has become necessary to develop crops with high water use efficiencies (WUEs) and drought tolerance using traditional breeding or genetic manipulation methods [1, 2]. Plants have a range of mechanisms to cope with drought stress at various levels, such as morphological, physiological, anatomical, tissue, cellular, and molecular levels. Generally, drought stress triggers signal transduction of the plant hormone abscisic acid (ABA). ABA is a key phytohormone involved

**Funding:** The author(s) received no specific funding for this work.

**Competing interests:** The authors have declared that no competing interests exist.

in stomatal closing to decrease transpiration. Drought also inhibits cell growth and photosynthesis, increases respiration and induces genes that respond to environmental stress [3, 4]. There is a significant difference in plant tolerance to drought stress relying on the duration and intensity of stress, the plant species and the growth stage. It has been reported that C4 plants are better adapted to drought areas than C3 plants and have better WUE for the following reasons: Kranz anatomy, having both phosphoenolpyruvate carboxylase (PEPC) and ribulose-1, 5-bisphosphate carboxylase/oxygenase (RUBISCO) pathways in mesophyll and bundle sheath that reduce photorespiration and increase photosynthetic efficiency, and a lower $CO^2$ saturation point for photosynthesis [5]. However, there is limited information comparing the response of plants C3 and C4 to drought stress, especially at the molecular level. Therefore, evaluating these plants in terms of their response to drought stress, identifying genes associated with drought stress, exploring their functional relationships to develop drought-tolerant cultivars can be critical.

Genome-wide analysis methods, such as RNA sequencing (RNA-seq) and microarrays, allow researchers to simultaneously study the expression patterns of thousands of genes under different stress conditions. Currently, there is a greater focus on RNA-seq data as studies have shown that it is more effective in identifying differentially expressed genes (DEGs) compared to microarray technology [6, 7]. DEGs are key indicators of plant response to different environmental conditions. By merging the results of multiple studies, researchers can increase the reliability of their findings and produce a more accurate set of DEGs. In addition, combining gene expression information across species can improve the ability to identify conserved gene sets that are key components of biological responses. Meta-analysis is a powerful strategy to take advantage of the potential of transcriptome studies. Meta-analysis is the use of statistical methods to analyze and combine the results of several independent but relevant studies [3, 6, 8]. So, in this study, an attempt has been made to reach a better conclusion about the drought response mechanism in C3 and C4 plants by combining the results of different RNA-seq studies using a meta-analysis approach.

## Methods and materials

### Plant materials and drought treatment

The seeds of sunflower and Artemisia (as C3 and C4 plants) were obtained from the seed and plant improvement institute, Karaj, I.R. Iran. The seeds were treated with 70% ethanol and 1% sodium hypochlorite, then washed by distilled water and germinated on moist filter paper in Petri-dish. The germinated seeds were transferred to pots filled with a 1:1 mixture of sand/soil and grown in three biological replicates in a greenhouse. The pots were irrigated to field capacity (FC) until seedlings reached the four-leaf stages. Then, the seedlings were divided into two groups: control (C) and water-stress (WS). For drought stress, watering for C pots was maintained at FC, and withheld for WS pots. The soil water content (SWC) and leaf relative water content (RWC) were monitored daily, and leaf material was sampled based on SWC reaching 50±5% and 30±5% of FC, as moderate and severe drought, respectively. The leaf material of C and WS plants frozen quickly in liquid nitrogen and stored at -80 ˚C.

### RNA-seq dataset collection and pre-processing

To investigate the C3 and C4 plant response to drought stress, NCBI Sequence Read Archive (SRA) (https://www.ncbi.nlm.nih.gov/sra) and EBI ArrayExpress database (https://www.ebi.ac.uk/arrayexpress/) were searched in the winter of 2020 using the following keywords: drought, drought stress, water stress, abiotic stress, and their combinations with an organism such as drought stress and *Zea mays* [organism]. Our considerations were, "The dataset should

be an RNA-seq gene expression profile, the data should be generated using the Illumina HiSeq platform, the samples should include both control and drought-treated groups, and the tissue type should be leaf". Finally, nine datasets including SRP071248, SRP042233, ERP107297, SRP045409, SRP101470, SRP110211, SRP135093, SRP106756, and SRP057095 were chosen, belonging to five different plant species: wheat, rice, barley, maize, and sorghum. RNA-seq raw reads (Fastq format) were downloaded from the European Nucleotide Archive database (https://www.ebi.ac.uk/ena/browser/home) (S1 Table).

The FastQC tool [9] was utilized to verify the quality of the raw data. Subsequently, the Trimmomatic tool [10] was employed to remove low-quality bases and adapter sequences. Following this, the resulting high-quality reads were mapped to the corresponding plant reference genome (http://ftp.ebi.ac.uk/ensemblgenomes/pub/release-49/plants/) using STAR software [11].

## Orthology definition and meta-analysis

In order to make a comparison of the transcriptional response among diverse species, it was necessary to identify orthologous genes. In this study, we considered Arabidopsis as a reference plant and the orthologous genes of each plant species based on Arabidopsis genes were obtained from the Ensembl and in the next steps the TAIR ID was used.

To remove batch effects across different datasets, the SVA package was utilized. Subsequently, a meta-analysis of C3 and C4 groups was carried out individually using the MetaDE R package. The gene expression profiles were merged by the MetaDE.merge Bioconductor package. Meta-analysis was limited to genes that are commonly found in all individual datasets. 30% unexpressed genes and 30% non-informative genes were filtered out. The number of permutation tests was set as 1000. Finally, to identify DEGs, Rank Prod method and Fisher method were used. Any genes that exhibited a false discovery rate (FDR) < 0.05 [12] were considered as DEGs and henceforth referred to as meta-DEGs.

A Venn diagram was created by the Venny 2.1.0 web-based software [13]. Gene expression values were determined by log ratio of means (ROM) through the following formula [14]:

$$y_{gn} = ln\left[\frac{\bar{r}_{gr}}{\bar{r}_{gs}}\right]$$

Where $y_{gn}$, $r_{gr}$ and $r_{gs}$, represent ROM, mean expression level of each gene in dataset.

## Gene ontology enrichment and pathway analysis of meta-DEGs

Gene ontology (GO) of meta-DEGs was conducted based on molecular function (MF), biological process (BP) and cellular components (CC) using SEA tool, agriGO (version 2.0) [15] with default settings (p-value, FDR <0.01). Therefore, the ontology gene of the meta-DEGs was compared to the ontology gene profile of the reference set. To identify the key pathways, a pathway analysis was carried out against the Kyoto Encyclopedia of Genes and Genomes (KEGG) database using the ShinyGO v0.61 tool [16].

## Protein–protein interaction network analysis

Protein-protein interaction (PPI) network analysis was executed using the STRING software (http://www.mybiosoftware.com/string-9-0-search-tool-retrieval-interacting-genesproteins.html). The resulting STRING network was visualized and analyzed by the Network Analyzer tool, which is available by default in Cytoscape software (version 3.7.2) [17]. Hub genes in the networks were also identified using the Cyto-Hubba plugin in Cytoscape software.

## Identification of transcription factor and miRNA families

Transcription factors (TFs) play an essential role in controlling the expression of genes in different environmental conditions. To identify TFs and *cis*-regulatory elements in the promoter regions of meta-DEGs, the Arabidopsis Gene Regulatory Information Server (AGRIS (http://arabidopsis.med.ohio-state.edu/)) was employed. Additionally, potential miRNAs were predicted by downloading meta-DEG sequences from the TAIR database (http://arabidopsis.org) and searching them against published miRNA sequences downloaded from miRBase through the psRNATarget server (http://plantgrn.noble.org/psRNATarget/).

## Validation by RT-qPCR

Total RNA was isolated from leaf tissue using DENAzist Column RNA Isolation Kit (DENAzist Asia Co., Mashhad, Iran) according to the manufacturer's instruction, followed by DNase treatment via *DNaseI*. cDNA was then synthesized using the YTA Reverse Transcriptase Kit as instructed by the manufacturer. The YTA SYBR Green PCR Master Mix was used for real-time quantitative PCR (RT-qPCR), and the relative expression level of each gene was normalized using the actin gene as a reference gene. The primer sequences for RT-qPCR can be found in S2 Table.

# Results

## Identification of DEGs using meta-analysis

To explore how C3 and C4 plants respond to drought stress and to identify which DEGs are shared or unique between the two groups, a meta-analysis was conducted on nine datasets. As a result, 693 and 528 meta-DEGs (adjusted p-value ≤ 0.05) were identified in C3 and C4 plants, respectively (S3 and S4 Tables). Of these DEGs, 41.6% and 23.3% were unique to C3 and C4 plants, respectively, while 35.1% (317 genes) were common in both groups. Among these common genes, 276 had similar expression patterns in both C3 and C4 plants, with 138 up-regulated and 138 down-regulated genes (Fig 1). A similar expression pattern in both C3

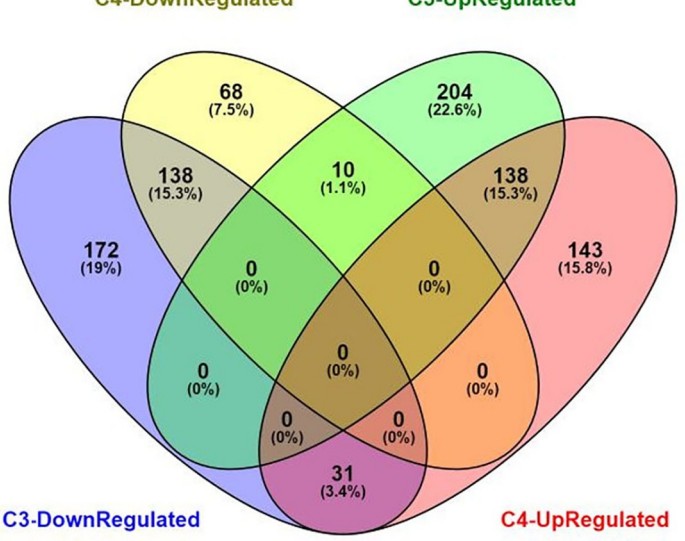

**Fig 1. Venn diagram of the number of unique and common differentially expressed genes (DEGs) found in C3 and C4 plant groups.**

and C4 plants suggested that they respond to stress in a similar way and implies their importance as key genes in stress response pathways. On the other hand, there were 31 genes that were down-regulated in C3 but up-regulated in C4, and 10 genes that were up-regulated in C3 but down-regulated in C4. The presence of genes with different expression patterns, as well as unique genes in each group, indicates that each group may have specific pathways for responding to drought stress.

## Identification of TFs and miRNAs

In this study, we identified 23 and 10 TFs from 14 and 6 TF families that respond to drought stress in C3 and C4 plants, respectively (Tables 1 and 2). The bHLH, AP2-EREBP, Homeobox, C2H2, C2C2-Dof, and NAC families were common in both groups of plants. However, the GRAS, WRKY, bZIP, G2-like, NLP, ABI3VP1, C2C2-CO-like, and TCP families were only detected in C3 plants. The Homeobox, AP2-EREBP, C2H2, bHLH, and NAC families had the highest numbers of genes.

Through analyzing down-regulated meta-DEGs, we identified 79 and 42 miRNAs belonging to 39 and 28 miRNA families that target 65 and 37 down-regulated genes involved in various biological processes in C3 and C4 plants, respectively (S5 and S6 Tables). On the other hand, the results showed that the families of ath-miR5021 with 19 targets in C3 plants and 10 targets in C4 plants, and ath-miR5658 with 9 and 5 targets in C3 and C4 plants respectively were the most abundant miRNA. The ath-miR156, ath-miR414, ath-miR168, and ath-miR5020a were the other miRNAs with the most targets (S1 Fig).

Four miRNAs, namely ath-miR5021, ath-miR5658, ath-miR172, and ath-miR5016, were found to target five genes encoding TFs in C3 plants. These genes included C2C2-CO-like

**Table 1. List of identified Transcription factors for C3 plants group.**

| TF Locus Id | Gene Name, Synonym | TF Family Name |
|---|---|---|
| At2g23340 | - | AP2-EREBP |
| At2g46680 | *ATHB-7, ATHB7* | Homeobox |
| At3g01470 | *ATHB-1, ATHB1, HAT5, HD-ZIP-1* | Homeobox |
| At3g18290 | *EMB2454* | C2H2 |
| At3g59060 | *PIF5, PIL6* | bHLH |
| At3g62420 | *ATBZIP53* | bZIP |
| At5g08790 | *anac081, ATAF2* | NAC |
| At5g09330 | *anac082* | NAC |
| At5g11060 | *KNAT4* | Homeobox |
| At5g16540 | *ZFN3* | C2H2 |
| At5g39660 | *CDF2* | C2C2-Dof |
| At5g41030 | - | TCP |
| At5g47220 | *ATERF-2, ATERF2, ERF2* | AP2-EREBP |
| At4g24020 | *NLP7* | NLP |
| At4g32010 | *HSI2-L1, HSL1, VAL2* | ABI3VP1 |
| At4g36870 | *BLH2, SAW1* | Homeobox |
| At4g38960 | - | C2C2-CO-like |
| At3g54990 | *SMZ* | AP2-EREBP |
| At1g50600 | *SCL5* | GRAS |
| At1g64625 | - | bHLH |
| At1g80840 | *ATWRKY40, WRKY40* | WRKY |
| At2g20570 | *GLK1, GPRI1* | G2-like |

**Table 2. List of identified Transcription factors for C4 plants group.**

| TF Locus Id | Gene Name, Synonym | TF Family Name |
|---|---|---|
| At2g22200 | - | AP2-EREBP |
| At4g36870 | BLH2, SAW1 | Homeobox |
| At2g46680 | ATHB-7, ATHB7 | Homeobox |
| At3g18290 | EMB2454 | C2H2 |
| At4g29930 | - | bHLH |
| At5g39660 | CDF2 | C2C2-Dof |
| At4g36920 | AP2, FL1, FLO2 | AP2-EREBP |
| At5g09330 | anac082 | NAC |
| At5g16470 | - | C2H2 |
| At5g11060 | KNAT4 | Homeobox |

(AT3G38960), Homeobox (AT4G36870), AP2/EREBP (At5g47220 and AT3G54990), and bHLH (AT3G64625). Additionally, the gene encoding C2H2 TF (AT3G18290), which was identified in both C3 and C4 groups, was targeted by three miRNAs, namely ath-miR5021, ath-miR2933, and ath-miR8173.

## GO enrichment and pathway analysis

To investigate the function of meta-DEGs in C3 and C4 plants, GO analysis was conducted. Results showed that out of 520 meta-DEGs in C4 plants, 420 genes were significantly enriched in 301 GO terms including 192 terms in the biological process category, 49 terms in the molecular function category, and 60 terms in cellular component category. In C3 plants, out of a total of 693 meta-DEGs, 566 genes were assigned to 355 GO terms in three categories of biological processes (226), molecular functions (57), and cellular components (72) (S7 and S8 Tables). Response to stress (GO:0006950), response to stimulus (GO:0050896), response to water deprivation (GO:0009414), response to hormone (GO:0009725), photosynthesis (GO:0015979) in the biological function category and catalytic activity (GO:0003824), binding (GO:0005488), oxido-reductase activity (GO:0016620), ion binding (GO:0016491), and antioxidant activity (GO:0016209) in the category of molecular function and in the cell components category also cell (GO:0005623), cell part (GO:0044464), organella (GO:0043226), and membrane (GO:0016020) were some of the most important common terms in both groups of plants (Fig 2).

The KEGG pathway analysis results revealed that the meta-DEGs in both C3 and C4 groups were significantly enriched in 52 and 48 pathways (FDR p-value≤0.05), respectively. The top 20 pathways based on fold enrichment and FDR are shown in Fig 3. Some common pathways in both groups of plants with the highest number of assigned meta-DEGs were biosynthesis of secondary metabolites (111 genes in C3 and 104 genes in C4), carbon metabolism (50 genes in C3 and 38 genes in C4), ribosome (27 genes in C3 and 19 genes in C4), amino acid biosynthesis (23 genes in C3 and 29 genes in C4), photosynthesis and carbon fixation (28 genes in C3 and 26 genes in C4), glyoxylate and dicarboxylate metabolism (22 genes in C3 and 13 genes in C4) and glycolysis/gluconeogenesis (19 genes in C3 and 18 genes in C4), starch and sucrose metabolism (14 genes in C3 and 19 genes in C4) and porphyrin and chlorophyll metabolism (12 genes in C3 and 12 genes in C4).

## PPI network analysis and identification of hub genes

Investigation and analysis of the interaction network of meta-DEGs was done using STRING. In C3 and C4 plants, networks were obtained with 693 and 528 nodes, 3339 and 2385 edges,

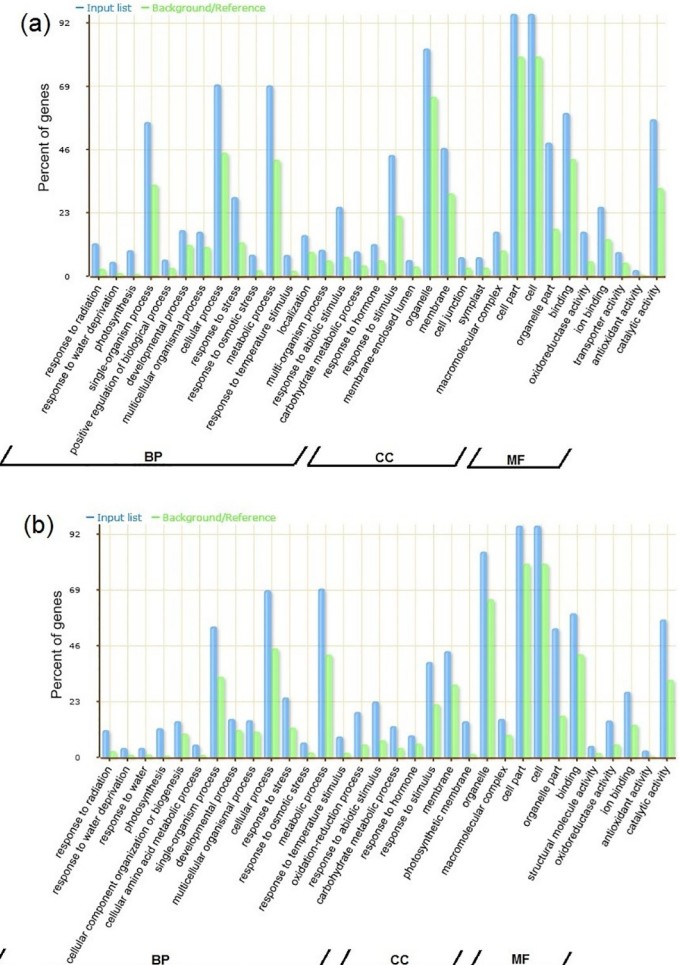

**Fig 2. Enriched GO terms corresponding to biological processes (BP), cellular components (CC) and molecular functions (MF) under drought stress in C3 (a) and C4 (b) plant groups.**

respectively, with an average node degree of 9.64 and 9.3 and PPI enrichment p-value<1.0e-16. The hub genes in the two networks were identified using the Cyto-hubba plugin in Cytoscape software. The top 50 nodes with the highest degree and central value of closeness were selected as hub genes, with 33 genes found to be common between the two groups (S2 and S3 Figs). These conserved genes were related to photosynthesis, antenna proteins, response to stimulus, response to abiotic stimulus, and ribosomal protein during the response to drought stress (Fig 4).

## Important pathways identified in response to drought stress

**Energy metabolism (photosynthetic pathway).** Among the identified meta-DEGs, 38 and 36 genes were related to energy metabolism in C3 and C4 plants, respectively, including carbon fixation pathway in photosynthetic organisms (22 and 19 genes, respectively, in C3 and C4), photosynthesis (6 and 7 genes, respectively) in C3 and C4) and antenna proteins (10 and 10 genes in C3 and C4, respectively). The findings revealed that the majority of the genes encoding enzymes involved in carbon fixation through the Calvin cycle were suppressed in

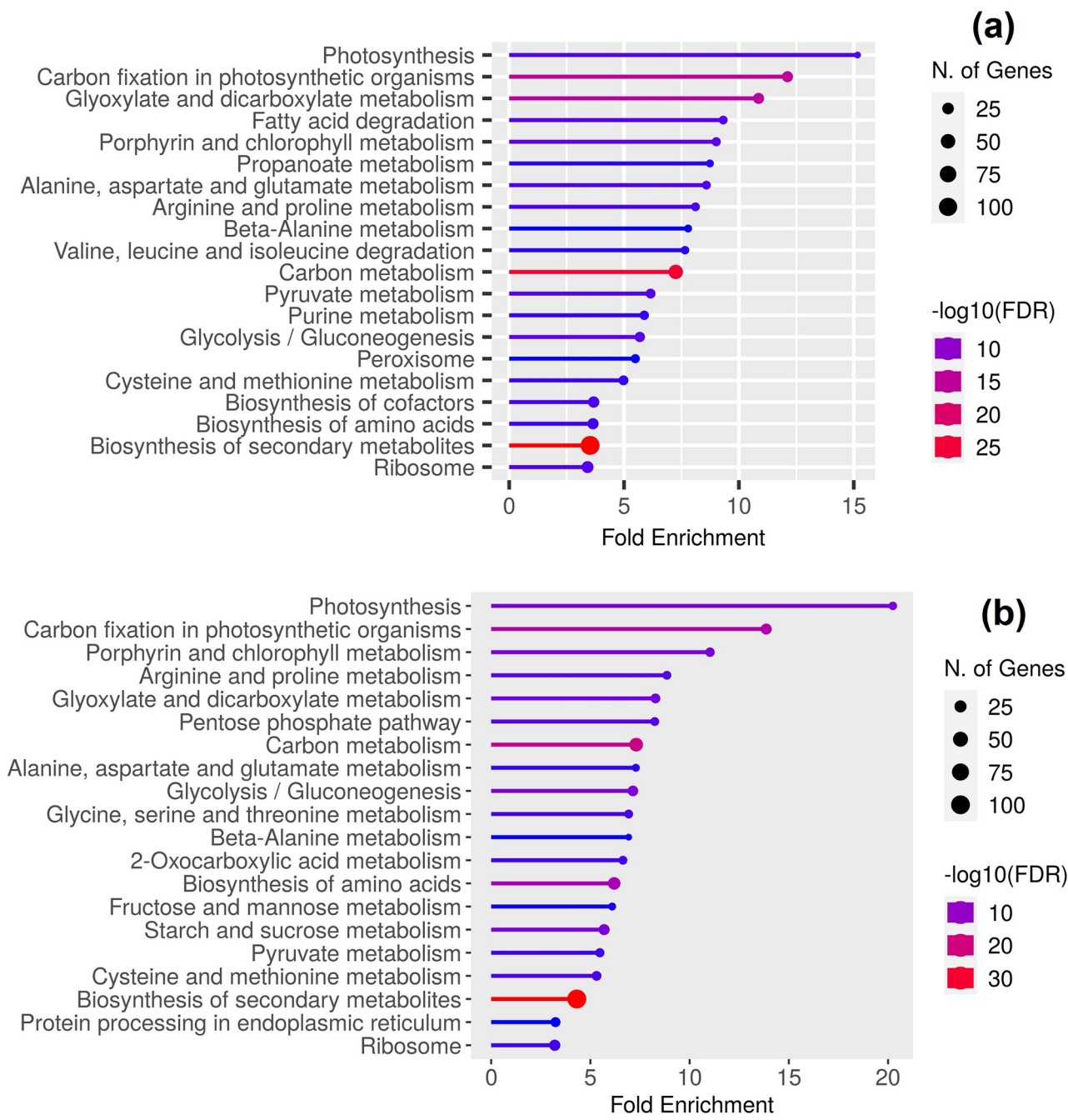

**Fig 3. KEGG pathway classification of the meta-DEGs in C3 (a) and C4 (b) plant groups under drought stress.**

both plant groups under drought stress. However, some genes exhibited diverse expression patterns, for instance, the gene encoding transketolase (AT3G60750) was up-regulated in C4 and down-regulated in C3, while the gene encoding glyceraldehyde-3-phosphate dehydrogenase C (AT3G04120) was down-regulated in C4 and up-regulated in C3.

The expression pattern of all the meta-DEGs associated with the photosynthesis pathway and antenna proteins was similar in both C3 and C4 plants, except for the light-harvesting

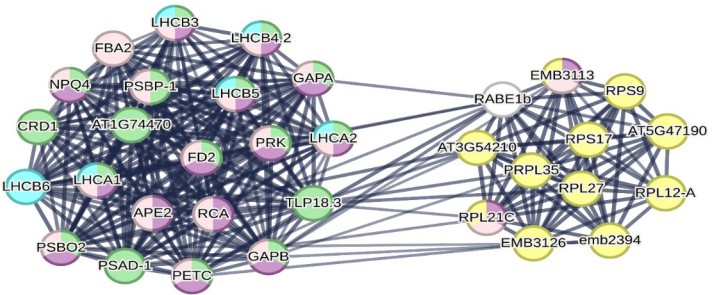

**Fig 4. Protein–protein interaction network of common hub genes in C3 and C4 plant groups.** Photosynthesis (green nodes), antenna proteins (blue nodes), response to stimulus (pink nodes), response to abiotic stimulus (purple nodes) and ribosomal protein (yellow nodes).

complex photosystem II subunit 6 (*LHCB6*; AT1G15820) and *petF* (AT1G60950) genes. These genes were up-regulated in C4 and down-regulated in C3 plants, as shown in Fig 5.

On the other hand, 12 meta-DEGs encoding several key enzymes involved in porphyrin and chlorophyll metabolism were also identified in each group of C3 and C4 plants. Nine genes were common in two groups of plants, including up-regulated genes, *pheophorbide a oxygenase* (AT3G44880), *magnesium dechelatase* (AT4G11911) and *chlorophyll(ide) b reductase* (*NYC1*; AT4G13250), and down-regulated genes *magnesium-chelatase subunit chlH* (*GUN5;* AT5G13630), *HEMB1* (AT1G69740), *chlorophyllide a oxygenase* (AT1G44446), *geranylgeranyl diphosphate reductase* (AT1G74470) and *Mg-protoporphyrin IX monomethyl ester oxidative cyclase* (AT3G56940). Gene *glutamate-1-semialdehyde-2, 1-aminomutase* (*GSA1;* AT5G63570) was also up-regulated in C3 plants and down-regulated in C4 plants. Three down-regulated genes *HEMA2* (AT1G09940), *ferrochelatase 2* (*FC2*; AT2G30390) and *urophorphyrin methylase 1* (*UPM1;* AT5G40850) were also detected only in C3 plants. The up-regulated gene *protochlorophyllide oxidoreductase A* (*PORA*; AT5G54190) and down-regulated genes *coproporphyrinogen III oxidase* (AT1G03475) and, *chlorophyll synthase* (*G4*; AT3G51820), were identified exclusively in C4 plants.

The hub gene analysis identified many common top-ranked genes based on the MCC method in both groups of plants. A significant proportion of these genes were related to the energy metabolism pathway. Specifically, four of the genes were involved in carbon fixation in photosynthetic organisms, and 12 genes were related to photosynthesis and antenna proteins, as follows: *petF* (*ferredoxin*), *PsbO* and *PSAD1* (Photosystem I reaction center subunit II), *PSBS* (*NPQ4*: Photosystem II 22 kDa protein), *petC* (Cytochrome b6-f complex iron-sulfur subunit), *LHCA1-2* (light-harvesting complex I chlorophyll a/b binding protein), and *LHCB3-6* (protein binding to chlorophyll ll a/b light-harvesting complex II). In addition, two genes related to porphyrin metabolism pathway, *geranylgeranyl diphosphate reductase* (AT1G74470) and *Mg-protoporphyrin IX monomethyl ester oxidative* cyclase (AT3G56940) were also determined as hub genes.

**Carbohydrate metabolism.** The findings from the meta-analysis showed alterations in the expression of genes involved in carbohydrate metabolism pathways in both C3 and C4 plants. The results suggested that the expression of some genes related to the sucrose cycle and starch degradation, galactose metabolism and tricarboxylic acid (TCA) cycle were up-regulated in both groups. However, the majority of the genes involved in the pentose phosphate pathway (PPP) were down-regulated in C3 plants, in contrast to C4 plants.

In this study, in both groups of C3 and C4 plants, genes encoding key enzymes in sucrose and starch metabolism such as sucrose phosphate synthase (*SPS*) and sucrose synthase (*SuS*),

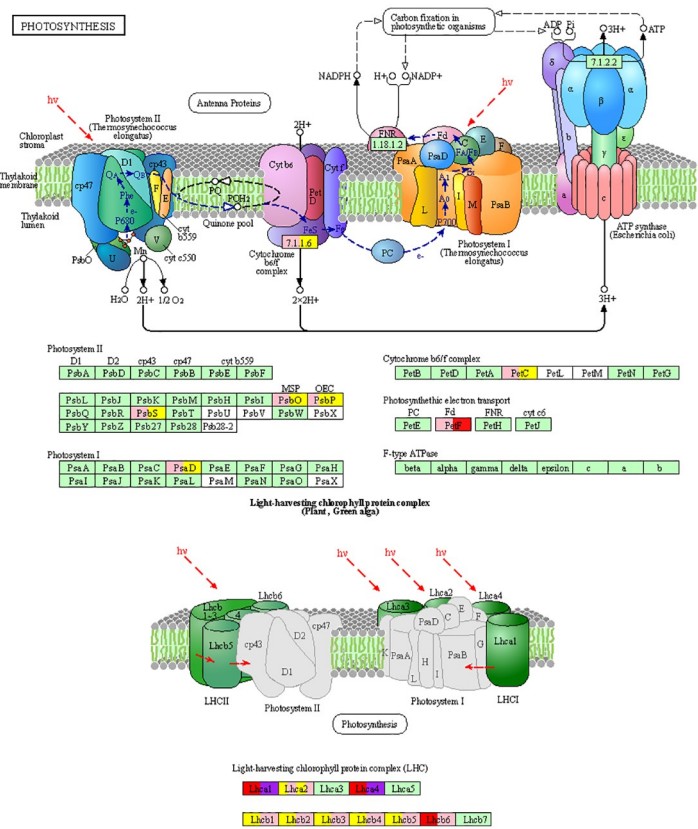

**Fig 5. Meta-DEGs related to the photosynthesis pathway and antenna proteins in C3 and C4 plant groups in response to drought stress.** Purple and pink color represent up and down-regulated genes in C3 group. Red and yellow color represent up and down-regulated genes in C4 group.

and starch synthase (*SS*), ADP-glucose-pyrophosphorylase, starch branching enzyme, as well as *beta-amylase (BAM)* and *isoamylase 3 (ISA3)* genes that have a role in starch degradation were identified. The investigation of the trehalose metabolism pathway revealed the identification of crucial genes including *trehalose-6-phosphate synthase 4 (TPS 4)* in both groups of plants and *trehalase* only in C3 plants. Under stress conditions, the expression of *TPS* gene, which is one of the two important enzymes involved in trehalose biosynthesis, was down-regulated, while *trehalase*, which is responsible for the breakdown of trehalose into glucose, was up-regulated.

Among the important genes up-regulated in the galactose pathway in both groups of plants, we can mention *alpha-galactosidase* genes (AT5G08380 (common in C3 and C4) and AT5G08370 (only in C3)) and *phosphoglucomutase* gene (AT1G70730 (only in C3)) which are involved in the synthesis and breakdown of glucose, as well as *raffinose synthase* genes (AT3G57520 (common in C3 and C4) and AT5G40390 (only in C4)) and *galactinol synthase* (AT2G47180 (only in C3)) which are involved in raffinose biosynthesis. Raffinose acts as an osmoprotectant in tolerance to abiotic stresses. In addition, transporters involved in sugar transport (section transporters) were also identified. Overall, these findings suggest that the activation of the sucrose cycle in the leaves of both C3 and C4 plants is a shared response mechanism to drought stress.

**Ribosome.** According to KEGG and network analysis, the "Ribosome" pathway was found to be significant. The study identified 27 genes associated with large (RPL) and small

(RPS) ribosomal subunit proteins in C3 plants and 19 genes in C4 plants. In C4 plants, all RP genes were observed to be down-regulated, whereas in C3 plants, 12 RP genes were up-regulated and 15 genes were down-regulated. Up-regulated genes *RPL3e*, *RPL4*, *RPL23A*, *RPL8*, *RPL7A*, *RPL10e*, *RPL6e*, *RPS3*, *RPS5e*, *RPS3A* and *RPS19* were detected only in C3 plants. 14 down-regulated genes including 10 RPL (RPL34, *RPL35*, *RPL1*, *RPL7/12*, *RPL18*, *RPL17*, *RPL21*, *RPL27*, *RPL6*, and *RPL19*) and four RPS (*RPS1*, *RPS17*, *RPS5*, and *RPS9*) were common in both groups. Five genes encoding *RPL24*, *RPL11*, *RPL29*, *RPL3* and *RPS13/18* were also detected exclusively in C4 plants.

**Amino acid metabolism.** The accumulation of amino acids is one of the plant responses to abiotic stress. Our study revealed that one of the important pathways with a considerable number of meta-DEGs was the amino acids biosynthesis pathway. Several key genes in the arginine and proline metabolism pathway were observed to be up-regulated. Furthermore, the expression of genes related to cysteine and methionine metabolism was altered in response to drought stress in leaves of both C3 and C4 plant groups. Transcripts encoding main genes involved in the methionine salvage pathway were also up-regulated. Besides, genes related to branched-chain amino acids (BCAAs) and lysine degradation were up-regulated in both C3 and C4 groups. However, genes related to the BCAAs biosynthesis pathway were only observed in the C4 plants group.

**Membrane transporters and channels.** In both C3 and C4 groups, several genes related to ion transporters and membrane channels were identified, including genes belonging to families of ATP-binding cassette (ABC), ATPase V type proton transporter (VHA), potassium transporter, and aquaporin. Also, we detected several transcripts encoding sugar, sugar alcohol, and amino acid transporters that regulate the movement and distribution of compatible solutes inside and between plant cells under stress. Including sucrose transporters/carriers (SUC), the SWEET sugar family of transporters, tonoplast monosaccharide transporters (TMT2), lysine-histidine transporter (LHT), proline transporter (PROT), and amino acid vacuolar transporters (AVT).

In both groups of plants, the up-regulated genes of *CAT2*, *SWEET* 6, and *ABCG22*, *ABCE6*, *ABCI8* from the ABC transporter family and three amino acid transporter genes *LHT*, *PROT3*, *AVT6B*, and *AVT1B* were common. Three genes *ABCG12 ABCG25* and *ERD6* and five genes *VHA-A*, *VHA-d2*, *VHA-H*, *VHA-D*, and *VHA-E1* from the VHA family, two genes belonging to the aquaporin family (*PIP1*, and *TIP1*) and two potassium transporter genes (*KUP8* and *KUP2*) were detected only in C3 group. Three genes *SUC4*, *TMT2*, and *GONST3* which belong to sugar and sugar alcohol transporters were also identified only in C4 group.

**Aldehyde dehydrogenase family member.** The results showed that drought stress induced the expression of genes encoding aldehyde dehydrogenase (ALDH) family member in both groups of C3 and C4 plants. Eight and five *ALDH* genes were detected in C3 and C4 plants, respectively. Five up-regulated genes *ALDH3I*, *ALDH5F*, *ALDH7B*, *ALDH10A* and *ALDH12A* were common in both groups. *ALDH11A*, *ALDH2B*, and *ALDH6B* were observed only in C3 plants. Among them, only *ALDH11A* was down-regulated.

## Validation by RT-qPCR

The meta-analysis results were validated by conducting RT-qPCR experiments on sunflower and Artemisia plants under drought stress. Four common genes in both groups of C3 and C4 plants were randomly chosen and their expression was compared between control and treatment plants (Fig 6). The results showed that the expression pattern of selected meta-DEGs under severe drought condition was the same as the meta-analysis results. These findings suggest that the meta-analysis results are reliable and accurate.

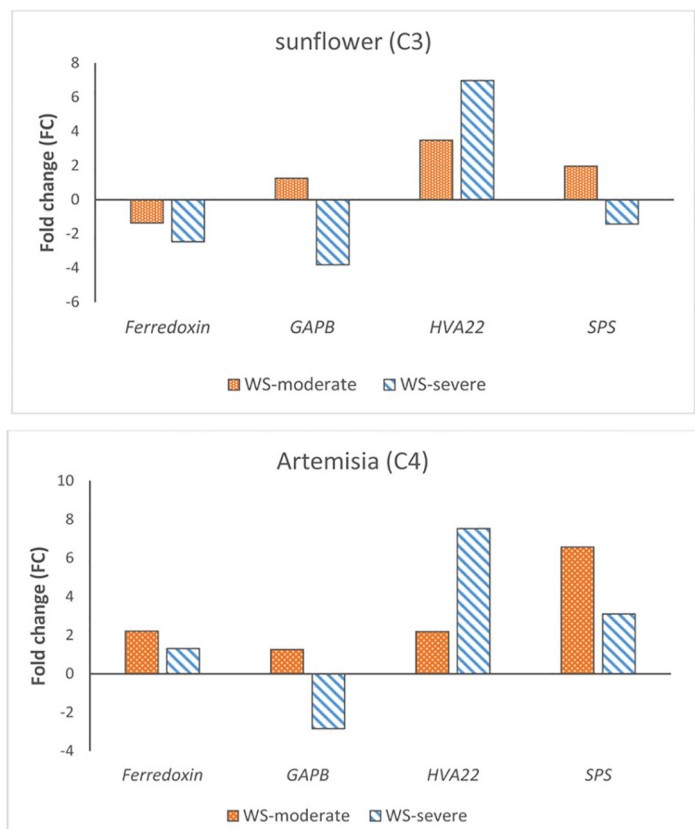

**Fig 6. Relative expression of four selected genes *Ferredoxin*, *GAPB*, *HVA22*, and *SPS*.**

## Discussion

### Identification of TFs and miRNAs

TFs play a crucial role in controlling gene expression in plants and have an impact on multiple functional processes such as response to environmental stimuli and hormones, cell growth and differentiation, and the development of organs. TFs work by binding to particular DNA sequences and interacting with various proteins within transcription complexes to regulate the expression of numerous genes [18]. Homeobox proteins, a group of TFs, are present in a wide range of organisms including invertebrates, vertebrates, fungi, and plants. In plants, HD-containing proteins are divided into 14 families such as HD-Zip WOX, BEL, and KNOX [19]. Different expression of homeobox genes in response to abiotic stresses has been reported in many plant species. For instance, an increase in *ATHB7* and *ATHB12* transcripts from the HD-Zip I group has been observed in plants subjected to drought stress [20]. HD-Zip genes as an important member of the homeobox family, participate in the response to diverse abiotic stresses in several plant species, including wheat, sesame, tea plant, foxtail millet, and potato [19]. Another largest group of TFs in plants is APETALA2/ethylene-responsive element binding proteins (AP2/EREBPs), which act as crucial regulators in various abiotic stress conditions such as drought, salinity, cold, and heat. Furthermore, AP2/EREBPs are involved in various hormone-related signal transduction pathways such as ethylene, ABA, cytokinin, jasmonate, and salicylic acid [21].

MicroRNAs modulate the response of plants to abiotic stress through post-transcriptional regulation [22]. Our research predicted several miRNAs from different families, including miR5021, miR5658, miR414, and miR156, in both C3 and C4 plant groups. MiR5021, which was found to have the highest number of target genes in our study, is a crucial miRNA involved in various pathways according to previous studies. For example, in Arabidopsis, certain miRNAs such as miR5021, miR158, and miR171 have been found to negatively regulate genes that contribute to cell growth, the ABA signaling pathway, and light-harvesting complex II (LHCII) [23]. Furthermore, Singh et al. [24] have proposed that miR5021, miR414, and miR156 may play a role in the secondary metabolite pathway, specifically in controlling the biosynthesis of essential oils by regulating the genes that encode enzymes involved in this process. In some plant species, miR156h has been observed to participate in various biological processes such as abiotic stress responses, hormonal balance, tissue growth, and cytoplasmic male-sterility [25]. The possible role of ath-miR5658 and ath-miR414 as the key regulators of post-transcriptional salt-stress responses has been pointed out [22]. Singh et al. [25] discovered that certain genes targeted by miRNAs, including miR5021, miR5658, miR408, and miR854, were involved in carbohydrate metabolism during abiotic stress. They also found that miR5021 targeted TFs such as WRKY and MYBs, which play regulatory roles in stress response. In the present study, C2C2-CO-like and C2H2 TFs are among the target genes of miR5021. It has been reported that C2H2 TFs are involved in Populus response to drought, salinity, and heat through different mechanisms [26]. Mun et al. [27] showed that drought stress significantly regulated several families of TFs, including AP2-EREBP, bHLH, MADS-box, WRKY, C2H2, C2C2-CO-like, C2C2-Dof, and homeobox.

Combinatorial regulation by miRNAs and TFs as the most important regulators of gene expression leads to an appropriate progression in biological events. Since many miRNAs are able to target TFs, they can be ideal candidates to investigate the interaction between gene expression networks and signaling pathways [18].

## Important pathways identified in response to drought stress

**Energy metabolism (photosynthetic pathway).**　　Photosynthesis is the most important metabolic process in plants which involves converting light energy into chemical energy [28]. This process occurs in two stages, with the first stage involving the absorption of light energy by antenna systems associated with photosystem I (PSI) and photosystem II (PSII) in higher plants [29]. LHC are responsible for several essential processes critical for plant growth, development, and response to abiotic stress [30]. In plants, LHCA1-LHCA6 genes encode LHCI, which mainly contains chlorophyll a, while LHCB1-LHCB7 genes encode PSII antennae [30]. The process of water splitting in the oxygen-evolving complex (OEC) leads to the production of $H^+$ ions in the thylakoid lumen. Electrons are transferred to PSI through the plastocyanin pool and cytochrome b6-f (Cytb6f) complex. Ferredoxin (FD) receives electrons resulting from water splitting in PSII, then ferredoxin $NADP^+$ reductase transfers the electron from the ferredoxin molecule to the nicotinamide adenine dinucleotide phosphate H (NADPH) molecule. ATP synthase transports $H^+$ ions back to the stroma, which converts the gradient energy of $H^+$ into chemical energy in the form of ATP [29].

Functional analysis of meta-DEGs in this study showed that genes related to photosynthesis and photosynthetic antenna proteins were significantly enriched. Photosynthesis-related genes such as PSI (*PSAD1*), PSII (*PsbO*, and *Psbs*), Cytb6f (*petC*), and photosynthetic electron transfer (*FD*) were determined as hub genes. *FD* was one of the genes with a differential expression pattern (up-regulated in C4 and down-regulated in C3) in two groups of plants. In addition to transferring electrons during photosynthesis, FD participates in reactive oxygen species (ROS)

scavenging by the reduced ascorbate and thus protects the photosynthetic system against photo-oxidative damage in plants under stress [31]. Down-regulation of *FD* gene expression in Arabidopsis (C3) and up-regulation in sorghum and maize (C4) have also been reported under drought stress [32–35]. *FD* up-regulation can lead to an increase in the stimulation of the electron transport cycle and the ability to dispose of excess electrons in drought stress conditions [34, 35].

LHCII-related genes, including *LHCB1-LHCB5*, were down-regulated in both groups of plants under drought stress. In contrast, only the LHCB6 coding gene showed up-regulation in the C4 group and down-regulation in the C3 group (Fig 5). This observation was similar to earlier research, which showed a decrease in the expression level of genes encoding LHC under abiotic stress, such as drought [28–30, 36, 37]. It is noteworthy that genes related to LHCI did not have the same expression pattern, as *LHCA1* and *LHCA4* genes were up-regulated, while *LHCA2* and *LHCA6* genes were down-regulated in both plant groups. Variation in *LHC* gene expression levels could be attributed to the chloroplast antenna's adaptation to excessive energy caused by high radiation or conditions limiting energy utilization such as cold or dehydration [36]. According to a prior investigation, LHCII assemblies and PSII-LH-CII super-complexes play a crucial part in protecting PSII from photo-damage during drought stress [37]. Therefore, based on our findings and previous research, LHC may play an important role in the response and adaptation of plants to drought stress. The reduction in the expression of genes related to photosynthesis and photosynthetic antenna proteins in our study implies that drought stress could affect the absorption and transmission of solar energy, as well as the efficiency of photosynthesis, in both C3 and C4 plant groups.

Chlorophyll is an important component of photosynthesis, and its content is significantly linked to the rate of photosynthesis and organic matter accumulation. A diverse range of enzymes catalyzes chlorophyll biosynthesis, and their activity reduction leads to its inhibition [38]. Earlier studies have revealed that under unfavorable conditions such as drought, the expression of enzymes involved in porphyrin and chlorophyll biosynthesis is down-regulated, which ultimately leads to a decrease in chlorophyll content [38–40].

Several genes, including *protoporphyrinogen oxidase* (PPO), *glutamyl-tRNA reductase*, and *magnesium chelatase* subunit, which are involved in porphyrin metabolism, are regulated under stressful conditions to enhance drought tolerance in plants [40]. In porphyrin biosynthesis, glutamate-1-semialdehyde and 5-aminolevulinic acid (ALA) are formed from glutamyl-tRNA, then catalyzed by glutamyl-tRNA reductase (encoded by the *HEMA* gene) and GSA, ultimately forming porphyrins through a variety of reactions. PPO as the last enzyme in the tetrapyrrole biosynthetic pathway is crucial for the biosynthesis of chlorophyll and heme. *ferrochelatase* (FC) and *magnesium chelatase* (composed of three subunits CHLD, CHLH, and CHLI) convert the PPO product (protoporphyrin IX (Proto IX)) into Fe-Proto IX and Mg-Proto IX, respectively [41].

In this study, drought treatment in C3 and C4 plants by down-regulation of genes encoding important enzymes of porphyrin and chlorophyll biosynthesis and up-regulation of genes involved in chlorophyll degradation may lead to inhibition of porphyrin and chlorophyll biosynthesis and ultimately reduces chlorophyll content. The decrease in chlorophyll biosynthesis pathway is a defense mechanism to avoid accumulation of singlet oxygen generating tetrapyrroles in the early stages of the stress response [42]. Porphyrin intermediates can produce phototoxic molecules such as singlet oxygen, so they are quickly utilized to form chlorophyll and/or heme. During stress, the reduction of chlorophyll production causes the accumulation of these intermediates along with ROS. It has been suggested that ROS, which are generated by porphyrin mediators, act as signaling molecules in the stress response pathway. Their levels increase, activating the stress response network, ultimately leading to the down-regulation of

genes involved in porphyrin metabolism [41]. On the other hand, the accumulation of ROS under stress leads to lipid peroxidation and consequently the destruction of chlorophyll. Reducing the rate of photosynthesis can also can cause a reduction in the abundance of proteins related to chlorophyll metabolism. Therefore, inhibiting porphyrin and chlorophyll metabolism may lead to more efficient energy saving in plants to defend against drought stress. Furthermore, the shift in leaf color from green to yellow due to chlorophyll reduction can increase radiation reflection, which serves as a protective mechanism for the photosynthetic system against stress [43].

In general, based on our results, the down-regulation of a wide range of genes involved in the photosynthesis pathway (light reactions and carbon fixation pathways), which leads to a lower photosynthesis and disturbance in the carbon fixation process, shows that the photosynthesis process in both groups of plants has been severely affected by drought.

**Carbohydrate metabolism.** Drought-induced water deficiency in plants results in decreased carbon fixation, attributed to stomatal closure in leaves and inhibition of photosynthetic activity, which in turn altered the carbohydrate metabolic balance. Additionally, plants accumulate a large amount of soluble carbohydrates such as glucose, fructose, sucrose, raffinose, mannitol, and pinitol under drought stress [44]. The expression of genes related to carbohydrate metabolism, including *SuS*, *SPS*, *TPS*, *trehalase*, *BAM*, *iso amylase*, *raffinose synthase* and *galactinol synthase*, was altered by drought stress in both plant groups in our study. SPS and SuS are key enzymes of sucrose metabolism in plants. SPS is required as the central enzyme in sucrose synthesis to produce sucrose-6-phosphate [45]. Drought stress affects *SPS* expression levels. Improvement of SPS activity changes the distribution of carbon and thus increases the content of soluble sugars and sucrose, which are important for regulating cell osmotic pressure to tolerate drought stress [40]. The reversible reaction between sucrose and fructose is catalyzed by SuS [45]. It seems that the up-regulation of *SuS* is activated in drought conditions for providing metabolic mediators, regulating osmotic potential, or respiration [46].

Starch is the main storage carbohydrate in plants, which plays a role in the response to abiotic stress in plants. Under drought conditions, the breakdown of starch in the chloroplast results in an increase in the levels of soluble sugars such as sucrose, glucose, and fructose [47]. Starch degradation is enhanced by increasing the expression of some key genes, including *ISA3*, *AMY3*, and *BAM9* [48]. The up-regulation of genes involved in the starch degradation pathway might help in allocating sugar energy during drought stress by promoting the conversion of starch into glucose [46]. The level of soluble sugar and sucrose increased while the level of starch decreased in soybean leaves under drought stress, possibly due to alterations in the activity of sugar metabolism enzymes and expression of genes such as *GmSuS*, *GmSPS*, *GmA-INV*, *GmC-INV*, *GmAMY3*, and *GmBAM1* [47].

Previous research has indicated the activation of genes involved in the galactose metabolism pathway, in addition to sucrose and starch metabolism [49]. In our study, an increase in the expression of genes involved in the galactose pathway was observed, such as the up-regulation of two raffinose synthase genes. Raffinose family oligosaccharides (RFOs) are known to accumulate in plants under abiotic stresses and act as osmolytes or antioxidants, as well as playing a role in carbon storage and transport [50]. Furthermore, galactinol synthase, which catalyzes the galactinol biosynthesis reaction, was only up-regulated in C3 plants. Galactinol is an important molecule in plant defense and acts as a galactosyl donor to generate larger RFOs [50].

We observed that drought stress caused changes in the expression of genes related to the glycolysis/glycogenesis pathway and the TCA cycle. Specifically, *citrate synthase* and *aconitase* were regulated in both groups of plants, while pyruvate dehydrogenase was only regulated in

C3 plants. Increased glycolysis can cause the accumulation of acetyl CoA in the TCA cycle and the production of more ATP, which could help plants withstand drought stress [51]. Therefore, increased glycolysis and TCA may be a strategy for energy supplying during the activation of the plant defense system against stress [52]. The PPP is another important mechanism of glucose degradation. Its main function is the production of substrates for the synthesis of other metabolites, particularly other monosaccharides [51]. PPP, as an important carbohydrate metabolic pathway, plays a key role in plant growth and stress response [53]. According to Shu et al [51], the down-regulation of PPP in rice under drought stress is thought to be related to the reduction of ribose, erythrose, shikimic acid, and arabinitol synthesis. This reduced synthesis may decrease unnecessary energy consumption under stress [51]. In contrast to C3 plants, up-regulation of important genes in PPP was observed in C4 plants, including the genes encoding glucose-6-phosphate dehydrogenase (*G6PDH*: AT5G40760), glucose-6-phosphate isomerase (AT4G24620) and transketolase (AT3G60750) (Fig 7). G6PDH is the essential enzyme responsible for the production of NADPH in the oxidative pentose phosphate pathway (OPPP) and has a vital function in modulating redox homeostasis and stress response [53]. OPPP produces a large amount of intermediate metabolites and NADPH, which are crucial for various metabolic processes such as fatty acid, nucleotide, and amino acid synthesis, carbon fixation, and nitrogen assimilation [54]. Furthermore, OPPP is a key mechanism in helping plants to cope with abiotic stress like salinity and drought [52, 53, 55]. It appears that NAD(P)H plays a crucial role in maintaining the activity of PSI when cells are under stress by serving as an electron source for PSI [56].

Transgenic soybean plants that overexpress *GmG6PDH2* exhibit greater resistance to salinity stress, likely due to the coordination with the redox states of ascorbic acid and glutathione pool to inhibit ROS production [53]. The induction of transketolase and NADPH indicates the operation of the PPP in plant cells under salt stress [57]. Under combined drought stress and heat shock, the expression of genes such as *G6PDH* increases the flow of sugars through these pathways, possibly to produce reducing energy, like NAD(P)H, when photosynthesis is not occurring [58].

**Ribosome.**   Plants use various mechanisms at the transcriptional and translational level to adapt to environmental changes. Ribosomes, due to their essential role in the process of translation and protein synthesis, can be very important in the plant's response to different environmental conditions [59]. According to previous research, many genes encoding ribosomal

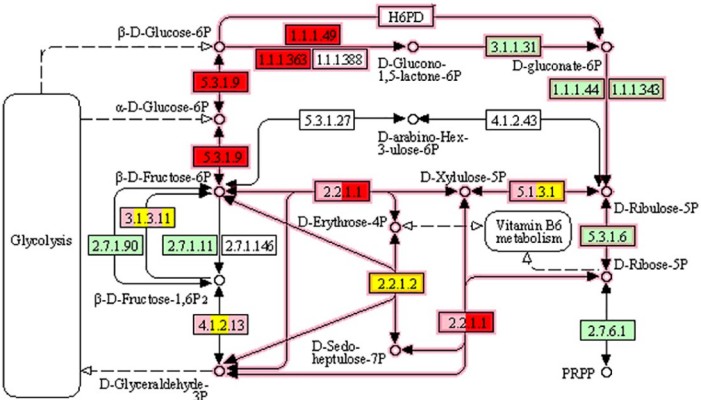

**Fig 7. Meta-DEGs related to the pentose phosphate pathway in C3 and C4 plant groups in response to drought stress.** Pink color represent down-regulated genes in C3 group. Red and yellow color represent up and down-regulated genes in C4 group.

proteins (RPL and RPS) were differentially regulated by various stress conditions which directly affect plant growth and transcriptional regulation of RP genes, ultimately leading to ribosome biogenesis [60]. In *Prosopis juliflora* various RP genes are commonly down-regulated under drought and salinity stress, indicating the start of the response to stress through a decrease in the rate of protein synthesis [61]. On the other hand, in rice some RP genes such as *RPL10*, *RPL6*, *RPL7*, *RPL23A*, *RPL24*, *RPL31* and *RPS10*, *RPS4* and *18RPS* were up-regulated when subjected to water stress [60]. Furthermore, studies on mutants of RP genes have also shown physiological defects in mutant plants. The reduction in RP function can affect protein synthesis, which may be one of the reasons for developmental abnormalities. These defects could suggest the extra-ribosomal and regulatory roles of RP [62]. The stress tolerance of transgenic plants can be improved through the expression of ribosome-related genes such as *RPL23A* [60].

The response of *Haloxylon salicornicum* to salinity stress involves an increase in the expression of proteins related to the ribosomal pathway, such as elongation factor-Tu (EF-Tu), which leads to an increase in protein synthesis [63]. The ability to renew proteins is one of the main protective mechanisms of plants in response to drought stress [59]. In our study, drought stress in C3 plants caused the up-regulation of genes encoding elongation factor EF-Tu proteins such as *S20e*, *S3e*, *L23Ae*, *L8e*, *L3e*, *L4e*, and *S2e*, but no significant change was observed in C4 plants. EF-Tu plays a crucial role in abiotic stress and has been reported to be up-regulated in plants under heat stress. The expression of maize EF-Tu gene in *Escherichia coli* found to induces heat tolerance. Knock-out EF-Tu mutants have shown lower EF-Tu protein levels and heat tolerance compared to wild-type plants. On the other hand, overexpression of an EF-Tu gene has improved heat tolerance [64]. Our study revealed a significant number of RP genes as hub genes (11 common genes (Fig 4), 14 genes in C3, and 12 genes in C4 (S2 and S3 Figs)), suggesting that RP genes play an important role in regulating the response to drought stress at the transcriptional level. Besides, the up-regulation of *EF-Tu* genes in C3 plants may indicate an increase in protein synthesis to restore damaged proteins during stress.

**Amino acid metabolism.** The initial response of plants to water deficiency is osmotic adjustment [51], which involves increasing the levels of certain amino acids to enhance stress tolerance. These amino acids serve as osmolytes, ROS scavengers, precursors of energy-related metabolites and signaling molecules. Studies on gene expression have demonstrated that various genes involved in amino acid metabolism, like proline, are regulated under drought stress [65]. In this study, the genes involved in amino acid metabolism were identified. Proline is one of the most important amino acids that accumulate in plants in response to stress and acts as an osmoprotectant and free radical scavenger [65]. It is mainly synthesized from glutamate in a sequential process involving γ-glutamyl kinase (γ-GK), pyrroline-5-carboxylate synthetase (P5CS), and P5C reductase (P5CR). Alternatively, it can be synthesized from ornithine through the action of ornithine-δ-aminotransferase (OAT), following the conversion of arginine to ornithine by arginase [66, 67]. Our research found that the expression of *P5CS*, *OAT*, and *arginase* were up-regulated in C3 and C4 plant groups. Previous studies have suggested that the up-regulation of *P5CS* gene expression leads to an increase in proline content in response to water stress [51]. The accumulation of proline under stress can be the result of stimulation of proline synthesis or the inhibition of proline degradation [67]. Our study also revealed that the proline dehydrogenase (*PDH*) gene (AT5G38710), which catabolizes proline to P5C in the process of proline degradation, was down-regulated exclusively in C3 group [68]. It has been reported that PDH activity decreases under stress [69]. The amino acids arginine and ornithine are precursors in the biosynthesis of polyamines (mainly spermidine, spermine and putrescine), which are ROS scavengers and can enhance plant tolerance to abiotic stresses. Therefore, it seems that arginine plays a role in plants response to abiotic stresses [70,

71]. Increased gene expression and protein accumulation of spermidine synthase (SPDS) and polyamine oxidase (PAO) induces polyamine biosynthesis in response to PEG stress [72]. We also observed the up-regulation of genes encoding SPDS (AT5G53120) in C3 plants and PAO (AT1G65840 and AT5G13700) in C4 plants. In maize under drought stress increase in the expression and activity of *PAO* in back-conversion of spermidine and Spermine to putrescine (acts as a protectant for the photosynthetic apparatus), as well as the elevated antioxidant activity have been reported. These results may contribute to the higher efficiency of the photosynthetic process and stress tolerance [73]. Therefore, it seems that the polyamine biosynthesis pathway may act through spermidine accumulation in C3 plants and putrescine in C4 plants in response to drought stress.

Glutamate, the main substrate of proline, can be produced through the saccharopine pathway as one of the lysine catabolism pathways. Abiotic stress or the induction of the aspartate pathway for lysine biosynthesis in the chloroplast can result in an increase in the cellular concentration of free lysine. Lysine biosynthesis in chloroplast requires a transporter such as LHT1 in order to transport the amino acid to the cytosol and other organelles to be metabolized effectively. LHT1 is induced under abiotic stress and can connect lysine biosynthesis to its catabolism. As lysine accumulates, the metabolic flux from lysine to α-aminoadipate increases via the saccharopine pathway [74]. Degradation of lysine through the saccharopine pathway is catalyzed by the bifunctional enzyme LKR/SDH and aminoadipic semialdehyde dehydrogenase (AASADH), which is encoded by the *ALDH7B1* gene in plants [65]. In both C3 and C4 plant groups, we observed an increase in expression of these genes involved in lysine metabolism in response to drought stress. Up-regulation of *LKR/SDH* and *AASADH* genes under drought stress in tolerant and susceptible sesame genotypes also has been reported [65]. The expression of *LKR/SDH* was up-regulated in response to stress-regulating hormones such as ABA and jasmonic acid [75]. In Arabidopsis and tobacco plants, ectopically expression of the *GmALDH7* gene enhanced tolerance to drought, salinity, and oxidative stress [76]. The up-regulation of genes involved in lysine degradation suggests that the activation of the saccharopin pathway under stress, through the production of osmoprotectants and the reduction of toxic aldehydes, contributes to stress tolerance in both plant groups.

In addition to proline, other amino acids especially branched chain amino acids (BCAAs) such as leucine, isoleucine, and valine also accumulate under abiotic stress. The accumulation level of BCAAs under stress are often higher or comparable to proline [77]. Water deficit has been shown to increase the levels of BCAAs in various species including Arabidopsis, barley, maize, and tomato [78]. While high-abundant amino acids like proline are synthesized in response to stress, low-abundant amino acids tend to accumulate due to increased protein turnover from proteolysis [79]. The accumulation of low-abundant BCAAs during stress due to increased protein degradation have been reported [77, 80]. The degradation of proteins increases in response to conditions that cause carbohydrate depletion, such as dehydration, salt stress, and extended periods of darkness. This process is essential to remove damaged proteins and to provide amino acids as a source of energy for the production of ATP, as well as to mobilize reduced nitrogen and sulfur [79].

The activation of the BCAA degradation pathway under stress conditions can provide an alternative source of respiratory substrates for the TCA cycle, as well as a detoxification mechanism by maintaining a free-branched amino acids pool at a compatible level with cellular homeostasis [81]. Our study has identified genes involved in BCAA degradation that are up-regulated under stress (Fig 8). Interestingly, we observed that genes in the BCAA biosynthesis pathway are exclusively down-regulated in C4 plants. Batista-Silva et al. [82] showed that during both drought and salinity stress, the biosynthesis pathways of low-abundant amino acids, including BCAAs and lysine, are down-regulated, while their degradation pathways are

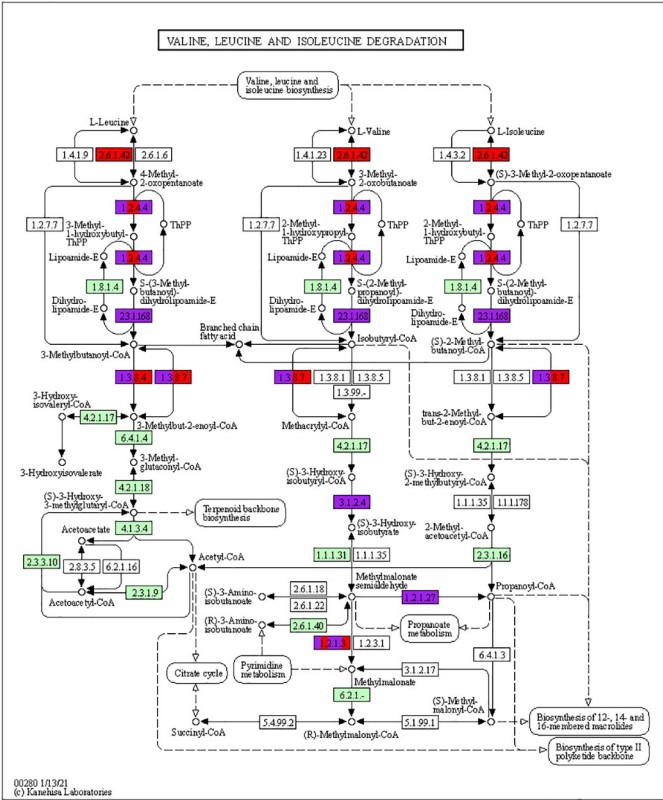

**Fig 8. Meta-DEGs related to branched chain amino acids (BCAAs; valine, leucine, and isoleucine) degradation pathway in C3 and C4 plant groups in response to drought stress.** Purple color represent up-regulated genes in C3 group. Red color represent up-regulated genes in C4 group.

strongly induced [82]. Degradation pathways of BCAA and lysine have previously been identified as essential factors for dehydration tolerance in Arabidopsis [83]. Wu et al. [84] also proved that BCAA degradation may be an important mechanism for drought resistance in sunflowers.

Our results can generally indicate the importance of the amino acid metabolism pathway in response to stress in both groups of plants, particularly the biosynthesis of proline and the degradation of BCAA and lysine. These pathways may help plants cope with stress by producing compatible osmolytes, supplying an alternative source of ATP, and detoxifying the harmful compounds. Another important pathway identified in our study is cysteine and methionine metabolism. Among the genes regulated in this pathway, we can mention *methionine synthase 3* and *S-adenosyl-L-homocysteine hydrolase*, which are key enzymes in the S-adenosyl-L-methionine cycle. Metabolites in this cycle have an important role in methylation of DNA, proteins, and other metabolites that control gene expression, cell wall metabolism, and polyamine and ethylene biosynthesis to enhance stress tolerance [85].

**Membrane transporters and channels.** Plants have developed complex strategies to adapt to the effects of changes in water status in the environment [86]. Membrane transport proteins are key targets for improving the efficiency of water and nutrient uptake and thus controlling drought tolerance in plants, they are also essential for transporting sucrose to where it is needed [86]. In this study, in addition to the identified important pathways, we

found expression of several genes encoding channels and membrane transporters such as ABC transporters, potassium transporter, VHAs, and sugar and amino acid transporters. ABC transporters are a type of primary active membrane transport protein found in the plasma membrane [87]. ABC proteins are categorized into several, ABCA–G and ABCI, subfamilies. ABC transporters translocate diverse molecules across a variety of biological membranes [88]. The ABCG subfamily as the largest known subfamily of ABC transporters in plants, can participate in various physiological processes [87]. This study identified three *ABCG* genes (*ABCG12*, *ABCG22* and *ABCG25*) that were up-regulated under drought stress. The role of ABCG22 and ABCG25 in stomatal regulation has been reported in Arabidopsis. *AtABCG22* and *AtABCG25* act in ABA influx into guard cells. Overexpression of *ABCG25* in Arabidopsis results in reduced water loss from leaves by transferring ABA to guard cells. *AtABCG22* mutant plants exhibit increased water transpiration and sensitivity to drought stress [89, 90].

VHAs are another group of up-regulated genes under drought stress in C3 group in our study. They are plant proton pumps which associated with oxidative phosphorylation and have a significant impact on plant growth and stress tolerance. The role of VHAs in response to abiotic stress have been widely investigated. For example, over-expression of the apple *VHA-A* gene in transgenic tobacco seedlings improved the activity of VHA and drought tolerance [91]. By overexpressing *GhVHA-A* in tobacco, it was discovered that this gene plays a crucial role in improving dehydration tolerance by enhancing osmotic adjustment and detoxifying ROS [92]. The differential expression of *VHA* genes in our study indicated their essential functions in response to stress in C3 plants, so drought may have a greater impact on oxidative phosphorylation in C3 plants than in C4.

Sugars are an important energy source and signaling molecule in plants [93]. Under drought stress, more efficient synthesis and movement of sugars within plant cells can be the main mechanism of drought tolerance due to the limitation of carbon absorption caused by stomatal closure and reduction of carbohydrates through respiratory processes [46]. The transport of sucrose through the apoplastic pathway in the plant phloem depends on the participation of SUC and SWEET families. The expression levels of genes encoding SUC and SWEET transporters are related to the capacity of sucrose transport [94]. Sugar transporters such as SWEET have a role in response to biotic and abiotic stresses. For example, *MdSWEET17* in transgenic tomatoes leads to higher tolerance to drought stress and more accumulation of fructose [93]. In our research, we observed the up-regulation of genes encoding various sugar transporters under drought stress, such as *SWEET6* (AT1G66770) in both groups of plants and *SUC4* (AT1G09960) only in C4 plants. The up-regulation of *SWEET* and *SUC* in soybean under drought stress can enhance the capacity of leaves to load sucrose and roots to discharge it [94]. In addition, increasing their expression in Arabidopsis leaves also increases carbon export from leaves to roots to maintain an efficient root system under stress [95]. Therefore, it is possible that the increase in the expression of sugar transporters in our study also indicates an increase in carbon flux from leaves to roots, especially in C4 plants due to their higher number of meta-DEGs encoding sugar transporters.

Several enzymes and transporters, which are involved in the biosynthesis of amino acids, are regulated under different environmental conditions. Amino acid transporters are the main mediators of nitrogen distribution in the plant and are necessary for maintaining growth and development. Environmental stresses such as salinity, light, and drought also affect the expression of amino acid transporters [96]. Our study showed an increase in the expression of amino acid transporters, including PROT3, LHT-like1, and AVTs, under drought stress. PROTs and LHTs are two types of amino acid transporters known to be involved in proline transport. [97]. PROTs are transporters that are mainly involved in the transport of proline, glycine, and γ-aminobutyric acid (GABA), while LHTs transport a wide range of neutral and acidic amino

acid substrates [96]. The AVTs are a group of proteins that are located on the vacuole membrane and are responsible for ATP-dependent transport of several amino acids, like proline, from the vacuole [98]. Previous studies on *AVT1B* knockdown mutants (avt1b-1 and avt1b-2) indicated lower glycine levels in the mutant plants compared to the control group. In addition, *AVT1B* expression is strongly suppressed in darkness. Therefore, AVT1B as a vacuolar glycine transporter acts in the negative regulation of glycine and can increase glycine storage [99].

**Aldehyde dehydrogenase family member.** Abiotic and biotic stresses cause the formation of ROS, which induce over-accumulation of aldehydes in cells [100]. Aldehydes are intermediate molecules in numerous cellular pathways, including amino acid, carbohydrate, protein, lipid, and steroid metabolism. However, excessive amounts of aldehydes have adverse effects on plant metabolism and can cause cellular damage [101]. To deal with harmful aldehydes, ALDHs oxidize a wide range of aldehyde molecules to their corresponding carboxylic acids [100]. Research conducted on transgenic Arabidopsis plants demonstrated that constitutive or stress-inducible expression of *ALDH3I1* and *ALDH7B4* genes causes high tolerance to osmotic and oxidative stress, which was associated with the decrease in the accumulation of ROS and malondialdehyde [101]. In our research, we observed that *ALDH3I* and *ALDH7B* were up-regulated in both the C3 and C4 groups under drought stress. We also identified other genes, such as *ALDH12A* and *ALDH2B* that showed increased expression. The increase in the expression of *StALDH12A1*, *StALDH7A1*, and *StALDH2B6*, in response to abiotic stresses, including dehydration, salinity, and heat has been reported [102]. ALDH12A1 plays a crucial role in preventing proline toxicity by degrading the toxic intermediate P5C [103]. Genes encoding ALDH5F1 and ALDH10 were also up-regulated in our study. ALDH5F1 catalyzes the conversion of succinic semialdehyde to GABA, a non-protein amino acid that accumulates in plants in response to abiotic stresses [104]. ALDH10 enzymes are related to polyamine catabolism and the biosynthesis of osmoprotectants. ALDH10 enzymes produce glycine betaine by oxidizing betaine aldehyde. Glycine betaine accumulates as an osmolyte in plants under osmotic stress [100]. These results may suggest that ALDHs may contribute to C3 and C4 plants coping with abiotic stresses through the elimination of toxic aldehydes by enhancing the antioxidant defense as well as their role in the metabolism of amino acids as an osmolyte.

The main limitation of this study was the number of species selected for meta-analysis due to the orthology definition. A limited species dataset may not fully cover the detailed growth and development stages challenged by drought. Therefore, we tried to select datasets related to important agricultural species. On the other hand, to validate the results of the meta-analysis through RT-qPCR and to generalize the results to C3 and C4 plants, different plant species (sunflower and Artemisia) were selected.

## Conclusion

Photosynthesis is one of the main plant processes that are affected by drought stress. Plants have different photosynthetic mechanisms including C3, C4, and CAM. This study aimed to investigate and compare the changes in the transcriptional profile of two groups of plants with different photosynthetic pathways (C3 and C4) in response to drought stress through RNA-seq meta-analysis. The accuracy of the meta-analysis results was confirmed by RT-qPCR. In total, 693 and 528 meta-DEGs were identified in C3 and C4 plants, respectively, and 276 of these genes had the preserved expression pattern in both groups. The identification of exclusive meta-DEGs of each group suggests that each plant group has a specific response to drought stress. The higher number of meta-DEGs in the C3 group implies that these plants are more affected by drought stress and have regulated the expression of more genes in different biological pathways to deal with stress. The analysis of putative miRNAs targeting DEGs

revealed that miR5021 and miR5658 were the most abundant groups. The functional enrichment analysis of these DEGs indicated that energy, carbohydrate, amino acid, and ribosome metabolism pathways were highly enriched in both C3 and C4 plant groups. Additionally, it was shown that ALDH and amino acid and sugar transporters also participate in stress response. Considering the expression changes observed in energy-related pathways, improving energy supply sources may be a promising strategy for enhancing drought tolerance in plants. The identification of common genes and pathways in both groups provides a comprehensive insight into the shared mechanism of stress response in plants.

## Supporting information

**S1 Fig. miRNAs and predicted targets in C3 and C4 plants under drought stress.**
(TIF)

**S2 Fig. Hub genes identified in in C3 plants group under drought stress.**
(TIF)

**S3 Fig. Hub genes identified in in C4 plants group under drought stress.**
(TIF)

**S1 Table. Characteristics of the individual RNA-Seq studies analyzed in this study.**
(XLSX)

**S2 Table. List of primers.**
(XLSX)

**S3 Table. List of the identified DEGs by meta-analysis in C3 plants group.**
(XLSX)

**S4 Table. List of the identified DEGs by meta-analysis in C4 plants group.**
(XLSX)

**S5 Table. List of identified miRNA for C3 plants group.**
(XLSX)

**S6 Table. List of identified miRNA for C4 plants group.**
(XLSX)

**S7 Table. Gene ontology enrichment analysis of the meta-DEGs in C3 plants group.**
(XLSX)

**S8 Table. Gene ontology enrichment analysis of the meta-DEGs in C4 plants group.**
(XLSX)

## Acknowledgments

We would like to thanks the Biotechnology Research Institute and high-performance computing (HPC) center of Shahrekord University for helping us to advance this study.

## Author Contributions

**Conceptualization:** Behrouz Shiran, Rudabeh Ravash.

**Data curation:** Shima Karami, Hossein Fallahi.

**Investigation:** Shima Karami.

**Methodology:** Rudabeh Ravash, Hossein Fallahi.

**Supervision:** Behrouz Shiran, Rudabeh Ravash, Hossein Fallahi.

**Validation:** Shima Karami, Behrouz Shiran.

**Writing – original draft:** Shima Karami.

**Writing – review & editing:** Behrouz Shiran, Rudabeh Ravash, Hossein Fallahi.

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
