## [Decision Letter · Decision Letter 0]

3 Apr 2023

PONE-D-23-06353A comprehensive analysis of transcriptomic data for comparison of plants with different photosynthetic pathways in response to drought stressPLOS ONE

Dear Dr. Shiran,

Thank you for submitting your manuscript to PLOS ONE. After careful consideration, we feel that it has merit but does not fully meet PLOS ONE’s publication criteria as it currently stands. Therefore, we invite you to submit a revised version of the manuscript that addresses the points raised during the review process.

We look forward to receiving your revised manuscript.

Kind regards,

Aimin Zhang, Ph.D.

Academic Editor

PLOS ONE

Journal Requirements:

2. Please include a copy of Table 1 and 2 which you refer to in your text on page 8.

Reviewers' comments:

Reviewer's Responses to Questions

**Comments to the Author**

1. Is the manuscript technically sound, and do the data support the conclusions?

Reviewer #1: Partly

Reviewer #2: Yes

2. Has the statistical analysis been performed appropriately and rigorously? 

Reviewer #1: Yes

Reviewer #2: Yes

3. Have the authors made all data underlying the findings in their manuscript fully available?

Reviewer #1: Yes

Reviewer #2: Yes

4. Is the manuscript presented in an intelligible fashion and written in standard English?

Reviewer #1: No

Reviewer #2: Yes

5. Review Comments to the Author

Reviewer #1: Authors need extensive English editing before reviewing process. I will review it after English editing. Most of the sentences are unusual and more than two meanings. Also, it has grammatical errors. In the current version, I can not reviewed it.

Reviewer #2: I suggest to address following two issues to further improve the quality of the manuscript.

1. As mentioned in “Materials and Methods” section, the datasets belonging to five plants wheat, Rice, barley, maize, and sorghum were selected and subjected to RNA-seq analysis including raw reads SRP071248, SRP042233, ERP107297, SRP045409, SRP101470, SRP110211, SRP135093, SRP106756, and SRP057095. The datasets are species limited and indeed not cover detailed growth and development stages challenged by drought. They are also not included for those deposited during past two years. Therefore, the authors should describe the shortages/shortcomings of this research in “Discussion” section.

2. The topics discussed in “Discussion” section need to be concentrated and focused on processes related to photosynthesis, especially on those impacting the behaviors of C3 and C4 plants treated by drought.

6. PLOS authors have the option to publish the peer review history of their article (what does this mean?). If published, this will include your full peer review and any attached files.

Reviewer #1: **Yes: **Aqarab Husnain Gondal

Reviewer #2: **Yes: **Kai Xiao

---

## [Author Response · Author response to Decision Letter 0]

8 May 2023

Dear Editor and Reviewers,

Thank you for your time and for providing valuable comments to improve our manuscript. Please find the answers to your questions and comments below. All parts that have been changed according to your comments are given in track changes in the revised manuscript. I hope that the revised file satisfies you.

Response to Editor

1. Please ensure that your manuscript meets PLOS ONE's style requirements

The revised manuscript was prepared based on PLOS ONE's style.

2. Please include a copy of Table 1 and 2 which you refer to in your text on page 8

We inserted Tables 1 and 2 in the revised manuscript.

Also a minor revision made in Figure 6 and Table S7-S8

Response to Reviewer #1 

1. Extensive English editing before reviewing process. 

We attempted to edit the entire text of the manuscript in order to make it more understandable. 

Response to Reviewer #2

Reviewer: I suggest to address following two issues to further improve the quality of the manuscript.

1. The authors should describe the shortages/shortcomings of this research in “Discussion” section.

According to the suggestion of the respected reviewer, the limitations of our research were mentioned at the end of the discussion section.

2. The topics discussed in “Discussion” section need to be concentrated and focused on processes related to photosynthesis, especially on those impacting the behaviors of C3 and C4 plants treated by drought.

We tried not to limit our study to the main difference between C3 and C4 plants (C3 and C4 cycle) because this leads us to focus only on the effect of drought stress on the activity of C3 and C4 cycle enzymes. Our aim is to obtain comprehensive information about the overall response occurring in the leaves of these plants, which also includes photosynthesis. Because the response of C3 and C4 plants to stress conditions is not only related to photosynthesis but may be related to extensive metabolic pathways. Also, in the energy metabolism section, we tried to explain common and different changes in the expression of genes related to photosynthesis in both groups of plants (Response of light reactions and carbon fixation pathways). Therefore, it seems that it would be better to refer to the photosynthetic pathway in the title of the energy metabolism section, hence this change has been applied in the new version. Furthermore, to complete the photosynthesis section some information has been added.

---

## [Decision Letter · Decision Letter 1]

2 Jun 2023

PONE-D-23-06353R1A comprehensive analysis of transcriptomic data for comparison of plants with different photosynthetic pathways in response to drought stressPLOS ONE

Dear Dr. Shiran,

Thank you for submitting your manuscript to PLOS ONE. After careful consideration, we feel that it has merit but does not fully meet PLOS ONE’s publication criteria as it currently stands. Therefore, we invite you to submit a revised version of the manuscript that addresses the points rai=============================

We look forward to receiving your revised manuscript.

Kind regards,

Aimin Zhang, Ph.D.

Academic Editor

PLOS ONE

Journal Requirements:

Reviewers' comments:

Reviewer's Responses to Questions

**Comments to the Author**

1. If the authors have adequately addressed your comments raised in a previous round of review and you feel that this manuscript is now acceptable for publication, you may indicate that here to bypass the “Comments to the Author” section, enter your conflict of interest statement in the “Confidential to Editor” section, and submit your "Accept" recommendation.

Reviewer #2: All comments have been addressed

2. Is the manuscript technically sound, and do the data support the conclusions?

Reviewer #2: Yes

3. Has the statistical analysis been performed appropriately and rigorously? 

Reviewer #2: Yes

4. Have the authors made all data underlying the findings in their manuscript fully available?

Reviewer #2: Yes

5. Is the manuscript presented in an intelligible fashion and written in standard English?

Reviewer #2: Yes

6. Review Comments to the Author

Reviewer #2: The authors have well addressed the concerns raised by the reviewers. A suite of minor errors are needed to be corrected before the publish in the journal.

Line 57: “CO2” needs to be corrected, in which “2” is lowercased.

Line 82 : “The Pots” needs to be corrected by “The pots”.

Line 90: “drought stress NCBI Sequence”: a comma needs to added ahead of “NCBI Sequence”.

Line 101: “S1 Table” should be “Table S1”.

Line 141: cis-regulatory elements: “cis” herein needs to be italic.

Line 154: “S2 Table” should be “Table S2”. Same as follow.

Line 163: “Fig 1” should be “Fig. 1”. Same as follows.\\

Line 174: “Table 1 and 2” should be “Tables 1 and 2”.

Lines 215-221: the capital letters needs to be lowercased. Same as in caption of Fig. 2.

Line 256: “Porphyrin” should be “porphyrin”.

Line 377: “are participate” should be “arparticipate”.

Line 389: “miR-158” should be “miR158”.

Line 403: “Mun et al. [27]findings showed” should be “Mun et al. [27 showed”.

Line 483: “can also can cause” should be “can also cause”.

Line 535: “as an important” should be “an important”.

7. PLOS authors have the option to publish the peer review history of their article (what does this mean?). If published, this will include your full peer review and any attached files.

Reviewer #2: **Yes: **Kai Xiao

---

## [Author Response · Author response to Decision Letter 1]

8 Jun 2023

Dear Editor and Reviewer

Thank you for your time and for providing valuable comments to improve our manuscript. We have made modifications to our updated manuscript point by point according to your comments. Our response follows (the comments are in italics).

We hope that the revised file satisfies you.

Response to Editor

Response: We checked all our references, the format of reference number 9 was modified and five references (31-35) were added in the previous revised manuscript (lines 431-437), one of which was forgotten (*), and this reference was also added in the new version. The number of references in the text and in the list of references has been updated, which can be traced in the Manuscript with Track Changes. On the other hand, for easier access to the references, the DOI of the articles was added.

9. Babraham Bioinformatics—FastQC A Quality Control tool for High Throughput Sequence Data [Internet]. [cited 2019 Feb 10]. Available from: http://www.bioinformatics.babraham.ac.uk/projects/fastqc/

31*. Lin YH, Huang LF, Hase T, Huang HE, Feng TY. Expression of plant ferredoxin-like protein (PFLP) enhances tolerance to heat stress in Arabidopsis thaliana. New biotechnology. 2015; 32(2): 235-242.‏ https://doi.org/10.1016/j.nbt.2014.12.001

32. Lehtimäki N, Lintala M, Allahverdiyeva Y, Aro EM, Mulo P. Drought stress-induced upregulation of components involved in ferredoxin-dependent cyclic electron transfer. J plant physiol. 2010;167(12): 1018-1022.‏ https://doi.org/10.1016/j.jplph.2010.02.006

33. Azzouz-Olden F, Hunt AG, Dinkins R. Transcriptome analysis of drought-tolerant sorghum genotype SC56 in response to water stress reveals an oxidative stress defense strategy. Mol Biol Rep. 2020;47: 3291-3303. https://doi.org/10.1007/s11033-020-05396-5

34. Fracasso A, Trindade LM, Amaducci S. Drought stress tolerance strategies revealed by RNA-Seq in two sorghum genotypes with contrasting WUE. BMC Plant Biol. 2016;16(1): 1-18. https://doi.org/10.1186/s12870-016-0800-x

35. Yang M, Geng M, Shen P, Chen X, Li Y, Wen X. Effect of post-silking drought stress on the expression profiles of genes involved in carbon and nitrogen metabolism during leaf senescence in maize (Zea mays L.). Plant Physiol Biochem. 2019;135: 304-309. https://doi.org/10.1016/j.plaphy.2018.12.025

Response to Reviewer 2

We would like to thank the reviewer for careful and thorough reading of this manuscript and for the thoughtful comments and constructive suggestions, which help to improve the quality of this manuscript. In addition, we appreciate the positive feedback from the reviewer. The responses are as below:

Reviewer #2: The authors have well addressed the concerns raised by the reviewers. A suite of minor errors are needed to be corrected before the publish in the journal.

Line 57: “CO2” needs to be corrected, in which “2” is lowercased.

Response: We have fixed the error.

Line 82 : “The Pots” needs to be corrected by “The pots”.

Response: We have fixed the error.

Line 90: “drought stress NCBI Sequence”: a comma needs to added ahead of “NCBI Sequence”.

Response: The correction has been made.

Line 101: “S1 Table” should be “Table S1”.

Response: We thank the reviewer for pointing this out. According to PLOS ONE's Supporting Information Citations "Format Supporting Information Citations as “S1 Fig”, “S1 Table”, etc.", we have cited this way.

Line 141: cis-regulatory elements: “cis” herein needs to be italic.

Response: The correction has been made.

Line 154: “S2 Table” should be “Table S2”. Same as follow.

Response: We thank the reviewer for pointing this out. According to PLOS ONE's Supporting Information Citations "Format Supporting Information Citations as S1 Fig, S1 Table, etc.", we have cited this way.

Line 163: “Fig 1” should be “Fig. 1”. Same as follows.\\

Response: We thank the reviewer for pointing this out. According to PLOS ONE's Figure Citations "Cite figures as Fig 1, Fig 2, etc.", we have cited this way.

Line 174: “Table 1 and 2” should be “Tables 1 and 2”.

Response: The correction has been made.

Lines 215-221: the capital letters needs to be lowercased. Same as in caption of Fig. 2.

Response: The correction has been made.

Line 256: “Porphyrin” should be “porphyrin”.

Response: The correction has been made. 

Line 377: “are participate” should be “arparticipate”.

Response: The correction has been made.

Line 389: “miR-158” should be “miR158”.

Response: We have fixed the error.

Line 403: “Mun et al. [27]findings showed” should be “Mun et al. [27 showed”.

Response: This observation is correct. We have changed.

Line 483: “can also can cause” should be “can also cause”.

Response: We have fixed the error.

Line 535: “as an important” should be “an important”.

Response: The correction has been made.

---

## [Editor Report · Decision Letter 2]

12 Jun 2023

A comprehensive analysis of transcriptomic data for comparison of plants with different photosynthetic pathways in response to drought stress

PONE-D-23-06353R2

Dear Dr. Shiran,

We’re pleased to inform you that your manuscript has been judged scientifically suitable for publication and will be formally accepted for publication once it meets all outstanding technical requirements.

Kind regards,

Aimin Zhang, Ph.D.

Academic Editor

PLOS ONE
---

## [Editor Report · Acceptance letter]

19 Jun 2023

PONE-D-23-06353R2 

A comprehensive analysis of transcriptomic data for comparison of plants with different photosynthetic pathways in response to drought stress 

Dear Dr. Shiran:

I'm pleased to inform you that your manuscript has been deemed suitable for publication in PLOS ONE. Congratulations! Your manuscript is now with our production department. 

Kind regards, 

on behalf of

Prof. Aimin Zhang 

Academic Editor

PLOS ONE